# Associations of Obstructive Sleep Apnea, Obestatin, Leptin, and Ghrelin with Gastroesophageal Reflux

**DOI:** 10.3390/jcm10215195

**Published:** 2021-11-07

**Authors:** Piotr Pardak, Rafał Filip, Jarosław Woliński, Maciej Krzaczek

**Affiliations:** 1Department of Gastroenterology with IBD Unit, Kliniczny Szpital Wojewódzki nr 2 im. Św. Jadwigi Królowej Affiliated with the Medical College of Rzeszów University, University of Rzeszów, 35-301 Rzeszów, Poland; r.s.filip@wp.pl; 2Department of Internal Medicine, Medical College of Rzeszów University, University of Rzeszów, 35-959 Rzeszów, Poland; 3Department of Animal Physiology, The Kielanowski Institute of Animal Physiology & Nutrition, Polish Academy of Sciences, 05-110 Jabłonna, Poland; jarek.wolinski@gmail.com; 4Department of Gastroenterology and Endoscopy, ZOZ Gastromed, 20-582 Lublin, Poland; one.teu@onet.eu

**Keywords:** gastroesophageal reflux, sleep apnea, ghrelin, obestatin, leptin

## Abstract

Gastroesophageal reflux disease (GERD) is commonly observed in patients with obstructive sleep apnea (OSA). Hormonal disorders observed in OSA may be relevant in the development of GERD. The aim of the study was to assess the correlations between ghrelin, obestatin, leptin, and the intensity of GERD in patients with OSA. The study included 58 patients hospitalized due to clinical suspicion of sleep disorders during sleep. All patients underwent a sleep study, and blood samples were collected overnight for hormonal tests. Survey data concerning symptoms of GERD, gastroscopy, and esophageal pH monitoring results were included in the study. In patients with OSA, GERD was twice as common when compared to the group without OSA. Among subjects with severe sleep apnea (AHI > 30; *n* = 31; 53%), we observed lower ghrelin levels, especially in the second half of the night and in the morning (*p*_5.00_ = 0.0207; *p*_7.00_ = 0.0344); the presence of OSA had no effect on obestatin and leptin levels. No significant differences in hormonal levels were observed between the groups depending on the diagnosis of GERD. However, correlations of ghrelin levels with the severity of esophagitis, leptin and ghrelin levels with the severity of GERD symptoms, and leptin levels with lower esophageal pH were found. GERD is more frequent among patients with OSA. In both GERD and OSA, deviations were observed in the levels of ghrelin and leptin. However, our analysis demonstrates that the relationship between OSA and GERD does not result from these disorders.

## 1. Introduction

Gastroesophageal reflux disease (GERD) is commonly observed in patients with obstructive sleep apnea (OSA) [1,2,3,4,5]. This disease is defined as a condition in which the stomach contents flow back into the esophagus, causing clinical symptoms and leading to the development of complications. Gastroesophageal reflux disease is diagnosed based on the clinical presentation and supporting examinations such as esophageal impedance–pH monitoring and gastroscopy [6]. Obstructive sleep apnea is characterized by the presence of multiple obturations on the level of the upper airways and leads to apnea, waking up, and loss of effective sleep [7]. The main complaints reported by patients included chronic fatigue, drowsiness and fatigue after a night’s rest, morning headaches, nocturia, excessive sweating, concentration and memory disorders, decreased libido, and depressed mood [8,9,10,11,12]. Snoring, which often occurs in OSA, is an insensitive indicator of the disease; however, when observed with apnea, it is highly indicative of OSA [13]. Due to the heterogeneous course of the disease and the low sensitivity of symptoms in the prediction of OSA, not every patient is referred for sleep diagnostics [14]. Therefore, tools were developed to identify patients at high risk for OSA. Some questionnaires in use include the Epworth Sleepiness Scale (ESS), Berlin Questionnaire (BQ), and Stop-Bang and Bang Questionnaires [7,9,10,15]. Assessment of upper respiratory tract morphology is also performed using the Modified Mallampati Score (MMP) and Upper Airway Volume measurement in cone-beam computed tomography (CBCT) [16]. Moreover, devices that monitor respiratory parameters, most often blood saturation or airflow through the respiratory tract, are also used in screening [7,17]. As a result, the qualification for more advanced but difficult to access and more expensive diagnostic methods is more effective. Diagnosis of OSA is based on results of a polysomnography (PSG). In patients without serious comorbidities and with symptoms indicating an increased risk of OSA, a home sleep apnea test (HSAT) performed with a portable monitor (PM) can be used to diagnose OSA [7,15,17]. Polysomnography includes the use of electroencephalogram (EEG), electrooculogram (EOG], and electrocardiogram (ECG). Additionally, airflow, oxygen saturation, respiratory effort, chin electromyogram, and heart rate are monitored. Monitoring of body position and lower limb movements is also recommended. In the home sleep apnea test, PM monitors at least airflow, respiratory performance, and blood oxygenation. In cases of diagnostic difficulties during the HSAT, PSG is recommended [7]. Obstructive sleep apnea is accompanied by numerous diseases, the most important of which are those involving the circulatory system [18]. In OSA, arterial hypertension and arrhythmias are more common, and the risk of coronary artery disease, sudden cardiac death, and stroke is increased [19,20,21]. Combined with excessive sleepiness and decreased concentration, this leads to a greater occurrence of traffic accidents [8]. Sleep apnea leads to the development of insulin resistance and type 2 diabetes and can be a cause of depression [8,18,22]. Moreover, sleep disorders have a strong impact on gastrointestinal diseases [23]. There is a strong relationship between OSA and GERD. It is estimated that up to 40–60% of people with OSA also suffer from GERD, which is often resistant to treatment and presents intense nocturnal symptoms, further reducing the quality of sleep [3,24,25]. Sleep disorders worsen the course of peptic ulcer disease (PUD), irritable bowel syndrome (IBS), functional dyspepsia (FD), and inflammatory bowel disease (IBD) [23,26,27,28]. In addition, many gastrointestinal diseases influence the sleep–wake cycle and sleep quality, especially in people with liver disease accompanied by hepatic encephalopathy, exacerbation of Crohn’s disease, or in the course of digestive tract functional disorders (IBS, FD) [26,27,29,30,31,32].

Motility disorders of the lower esophageal sphincter play a crucial role in the development of GERD in OSA patients [33,34]. The lack of association of apnea with reflux episodes and the beneficial effect of CPAP (Continuous Positive Airway Pressure) therapy indicate that hypoxia, inflammation, or other hormonal disorders might have a role in the development of GERD [34,35,36]. This can potentially result from disturbances in ghrelin, obestatin, and leptin concentrations, which are observed in OSA. This acts to worsen esophageal motility, LES dysfunction, or decrease the rate of gastric emptying, which may increase the likelihood of developing GERD.

Ghrelin and obestatin are products of the GHRL gene, synthesized mainly within the gastrointestinal tract. Functions of ghrelin include the stimulation of appetite and improvement of gastrointestinal motility. Furthermore, ghrelin can reduce inflammatory processes in the body. The level of ghrelin has considerable daily variability, and its secretion into the blood is of a pulsatile character, associated with the consumption of meals and the sleep–wake rhythm [37,38,39,40,41]. The physiological role of obestatin has not yet been fully elucidated due to, among other things, difficulties with the identification of the receptor on which it acts. It is presumed that obestatin affects glucose and lipid metabolism, decreases the inflammatory state, and exerts a beneficial effect on the survival of many cell types through the regulation of proliferative processes and inhibition of apoptosis [42]. Leptin is a hormone that is synthesized in adipose tissue. Through the inhibition of appetite, increased metabolism, and reduction in the amount of fatty tissue, leptin acts to regulate body weight. Chronically high leptin levels in obesity result in decreased sensitivity and loss of appetite inhibition [43]. Results from studies regarding the levels of ghrelin, leptin, and obestatin in OSA differ. In the majority of studies, elevated ghrelin and leptin levels were observed; however, the conclusions were formulated based on morning measurements [44,45,46,47]. In two available reports assessing the daily profile in OSA patients, greater leptin levels were observed, while significant deviations with respect to ghrelin and obestatin were not seen [46,48]. In the few available studies evaluating ghrelin and leptin levels in GERD, divergent results have also been noted. In some of these studies, lower ghrelin levels were observed in GERD, along with a decrease in the number of reflux episodes after administration of ghrelin. However, in other studies, either no significant deviations were found, or there was a clear tendency towards greater ghrelin levels in GERD. [49,50,51,52,53,54]. In the case of leptin, the majority of studies have demonstrated its greater levels in GERD [51,55]; however, there are no known studies evaluating the levels of obestatin in GERD.

Conclusions from these observations differ. Therefore, in our study, our aim was to determine whether the influence of OSA on the concentrations of obestatin, ghrelin, and leptin is important in the development of GERD. As a result, we hope to shed some light on the pathomechanisms linking OSA to GERD and to identify potential novel therapies for GERD.

## 2. Materials and Methods

The aim of the study was to assess the correlations between ghrelin, obestatin, and leptin and the intensity of GERD in patients with OSA.

### 2.1. Study Group

The study included patients hospitalized due to clinical suspicion of sleep breathing disorders. The majority of patients presented with snoring and pauses in breathing during sleep, excessive daytime sleepiness and chronic fatigue, drug-resistant hypertension, and cardiac arrhythmias. The exclusion criteria were: taking medications that disturbed sleep (sedatives and hypnotics) and GERD assessment (proton pump inhibitors, histamine receptor blockers, alkali); central sleep apnea; previous significant gastrointestinal surgery (gastrectomy, bowel resection); hypothyroidism, exacerbation of heart failure or COPD; or a history of stroke. The study was approved by the Ethics Committee at the Institute of Rural Health (Decision No. 6/2014). All patients signed an informed consent form prior to participating in the study. The study group consisted of 46 patients with OSA, while the control group consisted of 12 subjects in whom OSA was excluded. Sleep apnea was diagnosed when the AHI (apnea/hypopnea index) was at least 15 or at least 5 if clinical symptoms of OSA were present [7]. Survey data were collected concerning the clinical symptoms of GERD, sleepiness was assessed according to the Epworth Sleepiness Scale (ESS) [5,7], and blood was collected during the night. If indicated, patients were referred to the hospital gastroenterology outpatient department. Results obtained in the course of further diagnostics (gastroscopy and 24 h pH-metry) were included in the analysis. The diagnosis of GERD was made using the Lyon consensus definitions [6].

### 2.2. Sleep Examination

The overnight sleep study was performed according to current guidelines [7]. Sleep assessment was carried out based on the polygraph test (Type III acc. to AASM) using Embletta MPR PG (formerly Embla; currently Natus; Pleasanton CA, USA). Twelve patients underwent polysomnography (Type II acc. to AASM) using EmblaS4500 devices (formerly Embla; currently Natus; Pleasanton CA, USA) and RemLogic diagnostic software (formerly Embla; currently Natus; Pleasanton CA, USA). Classification of respiratory events and assessment of sleep apnea severity were performed by a sleep physician using the standard criteria defined by the guidelines of the American Academy of Sleep Medicine [7].

### 2.3. Questionnaire for Assessment of GERD Complaints

The intensity of complaints related to GERD was assessed based on a modified version of the questionnaire, which was used in a previous study (available online at Appendix A) [56]. This modified questionnaire contained items concerning an overall assessment of the intensity of GERD-related complaints (within the range 0–10 points) and questions pertaining to typical symptoms of GERD, with a distinction between those occurring during the day (within the range 0–40 points) and those occurring at night (within the scope 0–12 points). Regarding daytime complaints, the questions concerned their intensity and the frequency of a burning sensation in the chest (after meals, while lying down, or bending over); occurrence of burping or the feeling of stomach contents backing up into the esophagus accompanied by a sour or bitter taste in the mouth; and the sensation of difficulty swallowing (dysphagia). With respect to nighttime symptoms, patients were asked about the occurrence of GERD-related problems while falling asleep or waking up and the frequency of a sour or bitter taste in the mouth after a night’s sleep. The frequency of symptoms was classified according to a point scale: 0—when the symptoms did not occur; 1—when the symptoms occurred once or twice during the last month; 2—when the symptoms occurred no more than once a week; 3—several times a week; 4—several times a day. The intensity of complaints was assessed as follows: 1 point when complaints were mild; 2 points when they were clearly experienced; 3 points when complaints were troublesome; 4 points when complaints were difficult to tolerate. All questionnaires were collected from the patients by the primary investigator (P.P).

### 2.4. Gastroscopy and 24 h pH-Metry

To be included into our analysis, gastroscopy and pH-metry must have been performed prior to introduction of OSA treatment. All gastroscopic examinations were performed by an experienced endoscopist (systems: Fujifilm, Japan; Pentax, Japan) (R.F). For the purpose of statistical analyses, the intensity of inflammatory changes in the esophagus was ascribed a numerical value as follows: 0 for a normal result and from 1 to 4 for the subsequent inflammatory grades of the esophagus evaluated according to the Los Angeles classification [6]. pH-metry was performed using the ComforTEC Plus- PHNS single-channel probe (Sandhill Scientific, Highlands Ranch, CO, USA) placed 5 cm above the lower esophageal sphincter and REF: Z07-2000-A, SN: H109007C recorder (Sandhill Scientific, USA). After the performance of automatic analysis, the record was assessed by an experienced physician (M.K).

### 2.5. Determination of Total Ghrelin, Leptin, and Obestatin Levels

Blood was collected through a catheter inserted into the peripheral vein on the day following the sleep study at the following times: 23:00, 01:00, 03:00, 05:00, and 07:00. Blood sampling was performed at the patient’s bedside and did not require waking up the patient. Patients completed supper at least 3 h prior to collecting the first sample, and the last blood sample was collected more than 60 min before breakfast. In one patient, blood was collected only at 23:00 because the patient did not consent to collection of subsequent samples. Blood was collected using EDTA tubes, then centrifuged for 20 min at 3000 revolutions/min and a temperature of 4 °C (centrifuge MPW 260 R with angular rotor, MPW MED. INSTRUMENTS, Poland). Blood serum was transferred into Eppendorf tubes, frozen, and stored at a temperature of −80 °C until the measurements were performed (P.P). Hormonal measurements were carried out using commercially available radioimmunoassays (RIA): total ghrelin—Ghrelin (Human) RIA Kit (EMD Millipore’s Corp. Inc., St. Louis, MI, USA); leptin—Leptin (Human) RIA Kit (EMD Millipore’s Corp. Inc., St. Louis, MI, USA); obestatin—RIA Obestatin (Human, Monkey) RIA Kit (Phoenix Pharmaceuticals, Inc., Burlingame, CA, USA) (J.W).

### 2.6. Statistical Analysis

Statistical analyses were performed using the software package Statistica (data analysis software system), version 13 (TIBCO Software Inc. 2017; Palo Alto, CA, USA). The normality of the distribution of laboratory measurements was tested using the Shapiro-Wilk test. Because most of the features did not have a normal distribution, positional statistics (median, interquartile range—IQR) were mainly used to describe the results. Normality of the distribution was not observed, and this required the use of nonparametric statistical tests. Correlations between variables were calculated by means of Spearman’s rank correlation coefficient, while in order to assess differences between the two independent groups of patients, the Mann-Whitney U test was applied.

## 3. Results

### 3.1. Characteristics of the Study Group

The study included 58 patients (48 males and 10 females) aged 34–75 (mean = 54.5; Me = 56; s = 11.1). The patients were divided into two groups: one with the diagnosis of OSA (N = 46, including 5 females) and a control group (N = 12, including 5 females). A majority of the patients had an excessive body weight: 44 were obese (75.9%), while 14 were overweight (24.1%). The mean BMI was 34.8 kg/m^2^, while BMI values remained within the range of 25.1–49.7 kg/m^2^. Table 1 presents the basic characteristics of the study population.

### 3.2. GERD and ESS Questionnaires, Outcomes of the Sleep Study and Hormonal Measurements

No differences in the intensity of reflux symptoms were observed between the study and control groups. Additionally, no difference was noted in the level of sleepiness when evaluated according to the ESS score. In the group with OSA, a significantly lower value of minimum saturation during sleep was observed (73.9% vs. 85.1%; *p* < 0.001) along with a longer duration of snoring (Table 1). Table 2 presents the distribution of ghrelin, obestatin, and leptin levels. For technical reasons, the measurements of leptin and obestatin were performed in a smaller group of patients.

### 3.3. Gastroscopy and pH-Metry

Gastroscopy was performed in 27 patients. Inflammatory changes in the esophagus were observed in the majority of cases and were graded according to the Los Angeles classification. Grade A changes were seen in 22 patients, Grade B changes were seen in 7 patients, Grade C changes were seen in 1 patient, while no Grade D lesions were observed. Furthermore, in two patients, no inflammatory lesions were found [6]. PH-metry was performed in 23 patients; Table 3 presents the distribution of the variables. 

Correlations were investigated between the severity of esophageal inflammation, the symptoms of GERD, and results of pH-metry. Correlations near the level of statistical significance were found between the severity of esophageal inflammation, the symptoms of GERD, the DeMeester index, esophageal acid clearance time, and duration of the longest reflux episode (Table 4) [6].

### 3.4. OSA and Gastroesophageal Reflux Disease

In the OSA group, the diagnosis of GERD was confirmed in 35 (76.1%) patients, whereas in the control group, GERD was confirmed in 5 (38.5%) patients. Correlations were investigated between sleep parameters and the severity of reflux symptoms as assessed via questionnaires. The only significant correlation was observed in the study group and concerned the AHI values and symptoms of GERD during the day (r_s_ = 0.31; *p* = 0.0332); however, the correlation between AHI and symptoms of reflux during the night was near the level of statistical significance (r_s_ = 0.25; *p* = 0.0908). The correlation between the results of pH-metry and parameters of the sleep study was also examined. Correlations were found between the mean esophageal acid clearance time (after a reflux episode) and the mean (r_s_ = −0.55; *p* = 0.0084) and minimum (r_s_ = −0.41; *p* = 0.059) saturation at sleep, as well as between the duration of snoring and DeMeester score after meals (r_s_ = 0.39; *p* = 0.0657). No relationships were observed between sleep parameters and the severity of esophageal inflammatory lesions as assessed via gastroscopy.

### 3.5. OSA and Levels of Ghrelin, Leptin, and Obestatin

Hormonal levels were compared (ghrelin, leptin, and obestatin) between patients with AHI > 30 (31 patients, 53%) and patients with AHI < 30. The lowest ghrelin levels were noted in the group with severe sleep apnea, which were statistically significant for measurements performed at 05:00 and 07:00; however, no statistically significant differences were observed in the levels of obestatin and leptin (Figure 1; Table 5).

Analysis of the relationships between sleep parameters and the results of hormonal measurements indicated a correlation between the mean saturation during sleep and the level of obestatin at 03:00 (r_s_ = −0.29; *p* = 0.0631). In addition, together with an increase in the minimal saturation value during sleep, a tendency was observed towards greater levels of total ghrelin at 23:00 and 01:00 (r_s 23.00_ = 0.25; p_23.00_ = 0.0561; r_s 1.00_ = 0.26; *p*_1.00_ = 0.0509).

### 3.6. GERD and Levels of Ghrelin, Obestatin, and Leptin

No significant difference in the average level of ghrelin, obestatin, and leptin was observed between the group with GERD (N = 40) and the group without GERD (N = 18). However, a tendency towards greater values of ghrelin was found, together with greater severity of inflammatory changes in the esophagus. When divided into two groups, depending on the grade of esophageal lesions, significantly greater ghrelin values were noted in the group with more severe esophageal inflammatory changes (Figure 2; Table 6).

Analysis of relationships between the intensity of GERD clinical symptoms (based on questionnaires) and hormonal levels in the whole group did not show any significant differences, whereas, in the group of patients with GERD, significant correlations were found between the intensity of reflux symptoms and the levels of leptin and ghrelin (Table 7).

We analyzed the relationships between hormonal levels and parameters from pH-metry. Statistically significant correlations were observed between leptin levels and parameters from pH-metry. Furthermore, we noted a significant correlation between mean esophageal pH (at nighttime and during the whole day) and the mean esophageal acid clearance time in a recumbent position (Table 8).

## 4. Discussion

Our study presents the incidence and characteristics of reflux disease in patients with sleep apnea. In addition, we examined the profile of ghrelin, leptin, and obestatin in GERD and OSA. Correlations were also investigated between the levels of ghrelin, leptin, and obestatin and the parameters of GERD and OSA. Similar to other studies, we observed that GERD was more frequently diagnosed in the group with OSA compared to the control group [3,4,5,57]. Correlations were noted between OSA severity, the intensity of GERD clinical symptoms, and the minimum saturation values with prolonged esophageal clearance. This may indicate the importance of low saturation in the course of GERD and could explain the beneficial effect of CPAP therapy in patients without OSA who have been diagnosed with GERD [34,35,36]. A prolonged esophageal clearance and a more severe course of GERD in OSA were also observed in a study by Xiao et al. [58]. Similarly, an improvement in parameters of pH-metry was observed in several studies after the introduction of CPAP therapy [7,35,36]. In contrast, a study by Sabaté et al. did not observe a significant relationship between AHI and parameters of pH-metry, despite the more frequent occurrence of GERD in OSA [3].

In the presented study, parameters of pH-metry and clinical symptoms of GERD correlated with the severity of inflammatory lesions in the esophagus, whereas no correlation was found between OSA parameters and the severity of esophagitis. A study by Lee et al. reported more severe esophageal inflammatory lesions in OSA, which is in line with a study by Demeter et al. showing that the AHI value correlated with their severity [1,2].

After analyzing the ghrelin results, the group with severe obstructive apnea clearly differs from the others. In this group, lower levels of ghrelin were noteworthy, especially in the second part of the night and in the morning. In the available literature, conclusions regarding the level of ghrelin in patients with sleep apnea mainly involve morning measurements. A single study by Sanchez-de-la-Torre et al. evaluated the 24 h profile and observed no significant discrepancies in the ghrelin level in OSA [48]. A previous study by Liu et al. investigated the correlation between OSA and morning ghrelin levels. In line with our study, they noted lower ghrelin levels, especially in severe OSA [59]. In the majority of reports, the severity of OSA positively correlated with the level of ghrelin, and in studies by Garcia et al. and Chihara et al., effective CPAP therapy led to a decrease in ghrelin levels [44,45,60,61]. However, other studies showed no deviations in ghrelin levels in OSA, with CPAP therapy having no effect on its values [47,62].

In our presented study, no deviations in the level of leptin in OSA were observed, which is in line with the majority of previous reports [44,45,48,60,61,62]. In a comprehensive study by Arnardotiir et al., no deviations in morning leptin levels in OSA were found [63]. We have not found any reports where leptin levels are lower, whereas in several studies, including one which evaluated its daily profile, a greater level of leptin was observed in OSA [46,47]. It appears that elevated levels of leptin result from obesity, which frequently accompanies OSA, rather than the mechanisms related to sleep disorders. In our study, no deviations in the level of obestatin were found in the OSA group. Similarly, Zirlik et al. did not observe any significant deviations in the daily obestatin profile in OSA patients [46]. On the other hand, a study by Liu et al. showed that morning obestatin levels in the OSA group were lower; however, this observed difference did not reach the level of statistical significance [59].

In our study, no significant deviations were observed in the average levels of ghrelin, obestatin, and leptin in the group with a diagnosis of GERD. However, after analysis of GERD parameters (clinical symptoms, gastroscopic findings, and pH-metry), tendencies were observed towards greater levels of leptin and ghrelin, together with an increase in the severity of GERD. Similarly, greater levels of leptin in the group of patients with reflux esophagitis were observed by Nam et al., while Tomas et al. found correlations between leptin levels and the severity of GERD clinical symptoms [51,55]. Leptin does not exert an effect on the mechanisms associated with GERD, and therefore, it may be presumed that the observed correlation results from the fact that obese patients comprised a significant number in the examined group. The relationship between ghrelin and GERD is indicated by its action within the gastrointestinal tract and from observations in an animal model, as well as a study by Agrawal et al., where, after administration of ghrelin and capromorelin (a ghrelin agonist), a reduction in the number of reflux episodes was seen when compared to placebo [49,64]. To the best of our knowledge, there have been no reports regarding ghrelin levels in GERD. Moreover, in studies analyzing its morning levels, divergent results are encountered. In a study by Nishizawa et al. concerning functional dyspepsia, a tendency is clearly noted towards greater ghrelin values in the group with GERD [50]. In contrast, a study by Tseng et al. found no differences in ghrelin levels between the group with GERD and the control group [52]. Similarly, in a study by Thomas et al., no relationship was found between ghrelin levels and the severity of GERD symptoms; however, ghrelin levels correlated with an increased risk for the development of Barrett’s esophagus [51]. Additionally, Rubenstein et al. found a relationship between ghrelin values and an elevated risk for the development of Barrett’s esophagus; however, there was a negative correlation between ghrelin levels and GERD [53]. In another study, Shindo et al. also reported lower levels of ghrelin in GERD [54]. Discrepancies between the above-mentioned reports may indicate a complex relationship between ghrelin and GERD. Theoretically, high ghrelin levels may lead to GERD by increasing the secretion of gastric juice and decreasing its pH or may also result from esophageal wall lesions as a result of the inflammatory process, as in the case of IBD or celiac disease [40,65,66]. On the other hand, ghrelin stimulates stomach motility; thus, decreased levels of ghrelin prolong the duration of gastric emptying, which may be conducive to GERD. No deviations in obestatin levels were observed, and our study is the first to present its profile among patients with GERD.

### Strengths and Limitations of the Study

Our study has several limitations. Firstly, our study involved patients with a suspicion of OSA. Next, the control group was small and was not adjusted for BMI or distribution of adipose tissue. Additionally, the evaluation of OSA was performed on a different night than the collection of blood specimens. Finally, a gastroscopy was performed on some patients, while other patients did not undergo endoscopic examination. However, a strength of our study is that we assessed numerous objective parameters of GERD and OSA in a relatively large group of patients. Another advantage of this study is the performance of hormonal measurements in a nocturnal profile (when apnea and decreases in saturation occur) and the fact that each procedure was performed by the same investigator, which allows for greater repeatability and minimization of errors associated with subjective assessment. Although we did not compare our results with patients of normal BMI, we would like to emphasize the relatively large number of patients in our study group, as well as our measurement of hormonal profiles, as opposed to individual measurements of hormones.

## 5. Conclusions

The results of our study confirm the frequent occurrence of GERD among patients with OSA. In both GERD and OSA, deviations were observed in the levels of total ghrelin and leptin; however, analysis of the results did not indicate that the relationship between OSA and GERD results from these hormonal deviations.

## Figures and Tables

**Figure 1 jcm-10-05195-f001:**
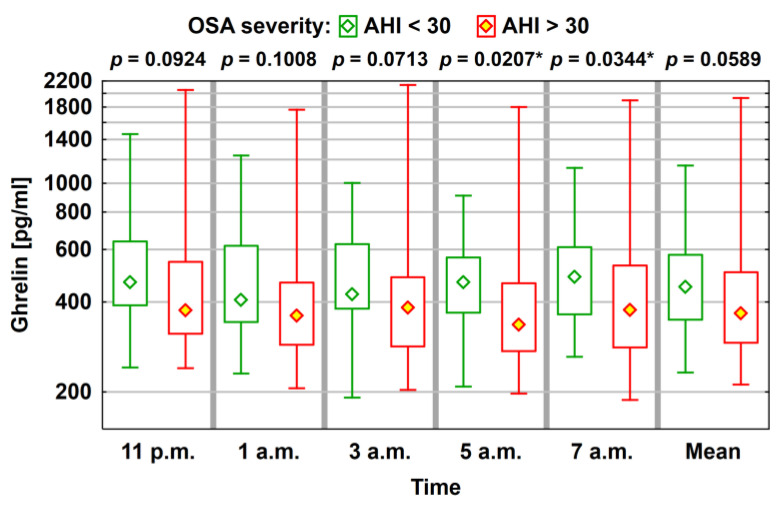
Average level (median) of ghrelin concentration (median) with a typical range of variability (lower and upper quartile) and the total range of variation (statistically significant correlations, where *p* < 0.05, are marked using *).

**Figure 2 jcm-10-05195-f002:**
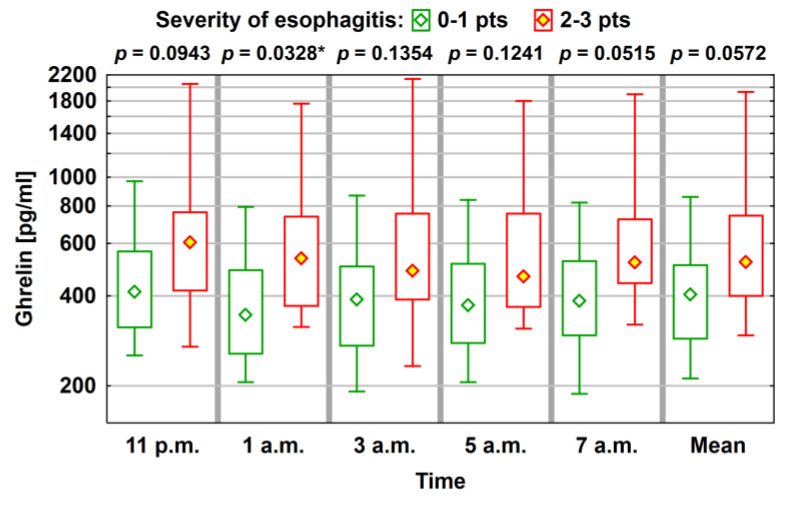
Average level (median) of ghrelin concentration with a typical range of variability (lower and upper quartile) and the total range of variation (statistically significant correlations, where *p* < 0.05, are marked using *).

**Table 1 jcm-10-05195-t001:** Basic characteristics of the study population, severity of GERD symptoms, and polysomnographic parameters’ distribution and comparison of variables between the test and control groups.

	OSA Group (*n* = 46) Mean ± Std. Dev.	Control Group (*n* = 12) Mean ± Std. Dev.	Both (*n* = 58) Mean ± Std. Dev.
Sex (M:F)	41:5	7:5	48:10
Age (year)	54.8 ± 10.6	56.5 ± 8.7	55.2 ± 10.2
Height (m)	173.9 ± 7.4	168.8 ± 13.2	172.8 ± 9.0
Weight (kg)	108.1 ± 21.1	92.5 ± 15.0	104.8 ± 20.8
BMI (kg/m^2^)	35.7 ± 6.5	32.7 ± 5.5	35.1 ± 6.4
Waist circumference	116.7 ± 13.4	111.0 ± 9.0	115.5 ± 12.8
GERD symptoms day (*p* = 0.3895)	11.4 ± 9.3	14.4 ± 9.2	12.1 ± 9.2
GERD symptoms night (*p* = 0.8637)	2.8 ± 3.0	2.7 ± 2.4	2.8 ± 2.9
GERD symptoms overall (*p* = 0.4416)	14.3 ± 11.7	17.1 ± 11.4	14.9 ± 11.6
ESS (*p* = 0.9209)	10.7 ± 5.2	11.3 ± 4.5	10.8 ± 5.0
AHI (*p* = x)	45.9 ± 25.8	6.3 ± 4.2	37.7 ± 28.1
Snoring time (*p* = 0.0744)	15.0 ± 16.3	7.8 ± 13.1	13.5 ± 15.9
SpO_2_ mean (*p* = 0.1693)	90.6 ± 5.2	93.0 ± 1.7	91.1 ± 4.7
SpO_2_ lowest (*p* < 0.001)	73.9 ± 12.1	85.4 ± 4.9	76.3 ± 11.9

**Table 2 jcm-10-05195-t002:** Distribution of ghrelin, leptin and, obestatin in the study group.

Laboratory Parameters	*n*	Mean	Median	IQR	Min	Max
Ghrelin at 23 (pg/mL)	58	509.1	445.2	244.9	240.2	2050.9
Ghrelin at 1 (pg/mL)	57	466.1	390.9	202.9	205.7	1762.4
Ghrelin at 3 (pg/mL)	57	476.0	416.3	193.6	191.5	2133.1
Ghrelin at 5 (pg/mL)	57	464.4	421.4	214.5	197.4	1798.4
Ghrelin at 7 (pg/mL)	57	490.8	431.1	247.9	188.0	1895.4
Ghrelin (mean) (pg/mL)	58	478.1	428.8	231.1	211.5	1928.0
Leptin at 23 (ng/mL)	40	30.2	27.9	12.2	17.9	48.7
Leptin at 1 (ng/mL)	41	29.1	26.5	10.7	16.5	43.2
Leptin at 3 (ng/mL)	41	29.5	27.5	10.8	15.9	47.7
Leptin at 5 (ng/mL)	40	28.9	28.0	9.6	16.0	44.2
Leptin at 7 (ng/mL)	41	28.7	28.1	10.7	16.2	44.1
Leptin (mean) (ng/mL)	42	29.2	27	10.3	17	45
Obestatin at 23 (pg/mL)	40	58.6	57.1	4.8	51.3	72.3
Obestatin at 1 (pg/mL)	41	67.9	66.4	8.0	55.4	89.4
Obestatin at 3 (pg/mL)	41	66.2	65.7	9.6	53.5	90.3
Obestatin at 5 (pg/mL)	39	59.8	59.7	5.6	50.9	70.2
Obestatin at 7 (pg/mL)	40	58.7	58.8	5.3	50.9	70.4
Obestatin (mean) (pg/mL)	42	64.1	61.9	5.5	53.9	98.2

**Table 3 jcm-10-05195-t003:** Distribution of esophageal pH monitoring results.

Parameters from pH-Metry	*n*	Mean	Me	IQR	Min	Max
De Meester index	23	27.8	15.1	30.2	1.7	136.4
De Meester index (post-meal)	23	9.8	7.5	10.5	1.5	37
GER episodes—recumbent	22	34.5	10	37.0	0	250
GER episodes—24 h	22	136.1	107	159.0	7	570
Mean pH (night)	23	6.4	6.5	0.9	4.3	8.1
Mean pH—24 h	23	6.2	6.2	0.8	5.0	7.2
Esophageal clearance time	22	37.0	35.5	24.0	8	91
Esophageal clearance time—recumbent	22	31.7	22.5	35.0	0	138
Longest GER episode	23	13.7	4.9		0.3	59.3

**Table 4 jcm-10-05195-t004:** Correlations of esophagitis severity with GERD symptoms and esophageal pH monitoring parameters (statistically significant correlations, where *p* < 0.05, are marked using *).

GERD Symptoms and Parameters from pH-metry	The Severity of Inflammatory Changes in the Esophagus–Gastroscopy Assessment
GERD symptoms—day	0.42 (*p* = 0.0172 *)
GERD symptoms—night	0.07 (*p* = 0.7034)
GERD symptoms—overall	0.33 (*p* = 0.0692)
De Meester index	0.38 (*p* = 0.0721)
De Meester index (post-meal)	0.40 (*p* = 0.0580)
GER episodes—recumbent	0.11 (*p* = 0.6258)
GER episodes 24 h	0.28 (*p* = 0.1994)
Mean pH—night	−0.01 (*p* = 0.9559)
Mean pH—24 h	−0.33 (*p* = 0.1234)
Esophageal clearance time	0.37 (*p* = 0.0858)
Esophageal clearance time—recumbent	0.42 (*p* = 0.0528)
Longest GER episode	0.38 (*p* = 0.0700)

**Table 5 jcm-10-05195-t005:** Comparison of ghrelin, obestatin, and leptin levels depending on the severity of OSA (statistically significant correlations, where *p* < 0.05, are marked using *).

Laboratory Parameters	OSA Severity						*p*
	AHI ≤ 30			AHI > 30			
	Mean	Median	IQR	Mean	Median	IQR	
Ghrelin at 23 (pg/mL)	544.6	469.2	249.2	484.5	374.6	232.2	0.0924
Ghrelin at 1 (pg/mL)	509.2	431.7	273.7	432.7	359.5	176.9	0.1008
Ghrelin at 3 (pg/mL)	501.2	434.8	245.3	457.6	382.3	200.0	0.0713
Ghrelin at 5 (pg/mL)	496.6	482.5	195.0	440.2	336.1	188.1	0.0207 *
Ghrelin at 7 (pg/mL)	531.8	494.4	218.4	462.6	376.5	247.4	0.0344 *
Ghrelin (mean) (pg/mL)	508.7	463.8	232.1	455.5	366.7	211.2	0.0589
Leptin at 23 (ng/mL)	29.6	28.3	9.7	30.4	27.6	14.4	0.9445
Leptin at 1 (ng/mL)	29.2	26.5	9.1	28.8	26.1	11.3	0.7455
Leptin at 3 (ng/mL)	29.6	28.7	9.9	29.1	26.9	11.2	0.7049
Leptin at 5 (ng/mL)	28.7	28.1	8.9	28.9	27.1	9.3	0.8129
Leptin at 7 (ng/mL)	28.4	28.6	10.4	28.6	27.3	11.1	0.9464
Leptin (mean) (ng/mL)	29.1	28	10.3	29.0	27	11.3	0.8864
Obestatin at 23 (pg/mL)	58.4	58.9	4.7	58.3	56.7	4.4	0.5489
Obestatin at 1 (pg/mL)	68.5	67.6	6.0	67.1	66.4	9.1	0.5334
Obestatin at 3 (pg/mL)	66.4	66.6	9.8	65.9	64.3	9.6	0.8711
Obestatin at 5 (pg/mL)	59.8	59.6	5.7	59.7	59.6	5.2	0.8966
Obestatin at 7 (pg/mL)	58.0	57.6	5.5	59.0	59.1	6.2	0.6061
Obestatin (mean) (pg/mL)	62.0	61.6	5.4	65.5	62.1	5.9	0.3829

**Table 6 jcm-10-05195-t006:** Comparison of ghrelin, obestatin, and leptin levels depending on the severity of esophagitis (statistically significant correlations, where *p* < 0.05, are marked using *).

Laboratory Parameters	The Severity of Inflammatory Changes in the Esophagus–Gastroscopy Assessment						*p*
	0–1 pts (N = 24)			2–3 pts (N = 8)			
	Mean	Median	IQR	Mean	Median	IQR	
Ghrelin at 23 (pg/mL)	452.7	412.1	250.4	735.9	604.0	345.5	0.0943
Ghrelin at 1 (pg/mL)	391.2	345.0	231.1	670.1	534.6	366.2	0.0328 *
Ghrelin at 3 (pg/mL)	413.3	388.6	229.0	702.9	485.3	364.9	0.1354
Ghrelin at 5 (pg/mL)	404.1	372.4	234.4	660.5	464.5	387.4	0.1241
Ghrelin at 7 (pg/mL)	423.3	384.4	227.1	697.3	518.0	281.7	0.0515
Ghrelin (mean) (pg/mL)	416.9	404.1	220.1	693.3	518.5	342.0	0.0572
Leptin at 23 (ng/mL)	32.1	30.0	18.0	30.0	28.3	7.3	0.9635
Leptin at 1 (ng/mL)	29.6	26.3	10.8	28.7	26.9	6.0	0.9023
Leptin at 3 (ng/mL)	29.8	24.9	13.0	29.2	28.1	5.6	0.7120
Leptin at 5 (ng/mL)	29.8	28.7	12.8	28.8	28.1	4.3	0.9635
Leptin at 7 (ng/mL)	30.2	31.1	9.6	29.2	29.3	4.4	0.6977
Leptin (mean) (ng/mL)	30.1	26.5	13.4	29.3	28	5.5	0.7743
Obestatin at 23 (pg/mL)	59.0	56.6	7.0	59.6	60.2	5.6	0.6167
Obestatin at 1 (pg/mL)	71.0	69.5	9.0	68.6	68.9	3.2	0.6521
Obestatin at 3 (pg/mL)	67.6	66.6	8.7	65.7	66.4	7.3	0.8375
Obestatin at 5 (pg/mL)	60.9	59.6	9.6	60.0	60.0	0.4	0.7505
Obestatin at 7 (pg/mL)	59.8	59.3	8.4	58.6	59.8	3.1	0.8916
Obestatin (mean) (pg/mL)	65.7	62.5	9.2	62.5	62.9	2.0	0.8375

**Table 7 jcm-10-05195-t007:** Correlations between ghrelin, leptin, and obestatin levels and the severity of GERD symptoms—group diagnosed with GERD (statistically significant correlations, where *p* < 0.05, are marked using *).

Laboratory Parameters	Severity of GERD Symptoms		
	DAY	NIGHT	OVERALL
Ghrelin at 23 (pg/mL)	0.31 (*p* = 0.0537)	0.10 (*p* = 0.5263)	0.27 (*p* = 0.0902)
Ghrelin at 1 (pg/mL)	0.29 (*p* = 0.0742)	0.05 (*p* = 0.7569)	0.23 (*p* = 0.1453)
Ghrelin at 3 (pg/mL)	0.24 (*p* = 0.1334)	0.14 (*p* = 0.3824)	0.23 (*p* = 0.1592)
Ghrelin at 5 (pg/mL)	0.29 (*p* = 0.0655)	0.14 (*p* = 0.3938)	0.26 (*p* = 0.0985)
Ghrelin at 7 (pg/mL)	0.28 (*p* = 0.0838)	0.02 (*p* = 0.9086)	0.21 (*p* = 0.1858)
Ghrelin (mean) (pg/mL)	0.30 (*p* = 0.0634)	0.09 (*p* = 0.5712)	0.25 (*p* = 0.1156)
Leptin at 23 (ng/mL)	0.35 (*p* = 0.0758)	0.30 (*p* = 0.1336)	0.39 (*p* = 0.0467 *)
Leptin at 1 (ng/mL)	0.37 (*p* = 0.0517)	0.30 (*p* = 0.1275)	0.39 (*p* = 0.0384 *)
Leptin at 3 (ng/mL)	0.36 (*p* = 0.0589)	0.29 (*p* = 0.1305)	0.39 (*p* = 0.0390 *)
Leptin at 5 (ng/mL)	0.39 (*p* = 0.0423 *)	0.26 (*p* = 0.1878)	0.40 (*p* = 0.0390 *)
Leptin at 7 (ng/mL)	0.25 (*p* = 0.2039)	0.23 (*p* = 0.2377)	0.27 (*p* = 0.1656)
Leptin (mean) (ng/mL)	0.36 (*p* = 0.0519)	0.27 (*p* = 0.1553)	0.38 (*p* = 0.0412 *)
Obestatin at 23 (pg/mL)	−0.01 (*p* = 0.9521)	−0.11 (*p* = 0.5955)	−0.03 (*p* = 0.8924)
Obestatin at 1 (pg/mL)	−0.08 (*p* = 0.6852)	0.19 (*p* = 0.3307)	0.00 (*p* = 0.9922)
Obestatin at 3 (pg/mL)	0.00 (*p* = 0.9850)	0.01 (*p* = 0.9422)	−0.02 (*p* = 0.9254)
Obestatin at 5 (pg/mL)	0.00 (*p* = 0.9894)	−0.08 (*p* = 0.7048)	−0.04 (*p* = 0.8372)
Obestatin at 7 (pg/mL)	0.01 (*p* = 0.9569)	0.09 (*p* = 0.6627)	0.01 (*p* = 0.9721)
Obestatin (mean) (pg/mL)	0.11 (*p* = 0.5829)	0.08 (*p* = 0.6870)	0.13 (*p* = 0.5152)

**Table 8 jcm-10-05195-t008:** Correlations between ghrelin, leptin, and obestatin concentrations and selected parameters of esophageal pH monitoring (statistically significant correlations, where *p* < 0.05, are marked using *).

Laboratory Parameters	De Meester Index	De Meester Index (Post Meal)	GERD Episodes—Recumbent	GERD Episodes—24 h	Mean pH (Night)	Mean pH—24 h	Esophageal Clearance	Esophageal Clearance—Recumbent	Longest Episode of GERD
Ghrelin at 23 (pg/mL)	0.31	0.30	0.25	0.36	0.02	−0.31	0.07	0.37	0.17
Ghrelin at 1 (pg/mL)	0.32	0.42 *	0.14	0.33	0.04	−0.27	0.16	0.37	0.21
Ghrelin at 3 (pg/mL)	0.30	0.37	0.27	0.33	−0.04	−0.26	0.23	0.39	0.23
Ghrelin at 5 (pg/mL)	0.34	0.38	0.29	0.32	−0.13	−0.29	0.24	0.41	0.22
Ghrelin at 7 (pg/mL)	0.25	0.24	0.26	0.25	−0.18	−0.37	0.19	0.34	0.14
Ghrelin (mean) (pg/mL)	0.31	0.35	0.24	0.32	−0.07	−0.30	0.19	0.41	0.20
Leptin at 23 (ng/mL)	0.31	0.07	0.33	0.33	−0.67 *	−0.59 *	0.01	0.62 *	0.14
Leptin at 1 (ng/mL)	0.17	0.05	0.14	0.17	−0.41	−0.41	−0.26	0.41	−0.01
Leptin at 3 (ng/mL)	0.34	0.11	0.34	0.32	−0.56 *	−0.60 *	−0.13	0.52 *	0.12
Leptin at 5 (ng/mL)	0.25	0.08	0.24	0.29	−0.45	−0.55 *	−0.13	0.49	0.04
Leptin at 7 (ng/mL)	0.30	0.16	0.25	0.25	−0.62 *	−0.60 *	−0.11	0.53 *	0.10
Leptin (mean) (ng/mL)	0.30	0.12	0.27	0.28	−0.55 *	−0.59 *	−0.09	0.53 *	0.09
Obestatin at 23 (pg/mL)	0.18	0.36	0.26	0.09	−0.20	−0.16	−0.04	0.24	0.18
Obestatin at 1 (pg/mL)	−0.06	−0.05	0.32	−0.09	0.09	0.09	−0.32	−0.07	−0.21
Obestatin at 3 (pg/mL)	−0.06	0.26	−0.14	−0.11	0.24	0.15	0.19	0.09	0.07
Obestatin at 5 (pg/mL)	0.28	0.41	0.22	0.35	0.20	0.01	0.43	0.07	0.26
Obestatin at 7 (pg/mL)	0.09	0.16	0.03	−0.08	0.17	0.00	−0.05	0.01	−0.14
Obestatin (mean) (pg/mL)	−0.04	−0.06	0.17	−0.04	0.21	0.23	−0.25	−0.06	−0.27

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
