# Peer review of "Associations of Obstructive Sleep Apnea, Obestatin, Leptin, and Ghrelin with Gastroesophageal Reflux"

_jcm, 2021, doi:10.3390/jcm10215195_

Round 1

Reviewer 1 Report

This is a very well written and conducted study concerning joint GERD-OSA screening and evaluation. Topic is extremely important and up-to-date as most of OSA cases and comorbidities are commonly underdiagnosed in general population. Authors assess the correlations between ghrelin, obestatin, leptin, and the intensity of GERD in patients with OSA. An entire study, despite some of its' weaknesses - which authors pinpointed in relevant section - is a large leap towards better understanding of intricate interplay between OSA, GERD and hormone levels. Especially that evidence in these cases still remains contradictory. However, some remarks should be addressed prior to publication.

Introduction

An entire section is brief, concise, yet does not lead to a clear hypothesis - please formulate at the end of this section, with your study expectations

L40-41 - I feel that authors should mention a few more words prior to PSG which stands first in OSA diagnostics. Approx. one out of four snoring patients is OSA positive according to AHI; thus not every patient is getting PSG - and it requires screening tools (Berlin questionnaire, Stop-Bang, CBCT, MMP assessment) with subsequent referrals. I strongly suggest incorporating one or few references about OSA screening here e.g. https://www.mdpi.com/2076-3417/11/9/3764 -yet, there is a paper published recently discussing concerns between OSA and self-reported breathing disorders which is worth mentioning https://pubmed.ncbi.nlm.nih.gov/33004829/ also one short phrase about polygraphy and polysomnography should be also addressed here with relevant citation e.g. https://pubmed.ncbi.nlm.nih.gov/32418018/ - this will improve the way of thinking and clarify the way to hypothesis stated.

Materials and methods

L579-80 - these statement need relevant citation which is missing

Also there are no clear inclusion and exclusion criteria - please outline carefully, without 'e.g.'s

L80 - list medications which were allowed and the ones which were considered as an exclusion. This is too important in a study assessing hormone levels.

L81-82 - 'significant surgery' or ''other conditions' - this is quite vague and not precise, please clarify or remove if redundant,

L93-94 - manuacturer's data is missing - please provide

L96 - no 'standard criteria' in scientific world do apply - please clarify, with relevant citation(s) and/or re-formulate/paraphrase an entire statement.

L99 - this questionnaire should be also included in a supplement section

L117 - initials are needed as there is no clear information about principal investigator in contributions as no relevant section is present, authors should outline their study involvement at the end of the manuscript, according to MDPI publishing policy.

L121 - initials are needed, if one of the authors contributed

L128 - initials are needed, if one of the authors contributed

L136 - provide centrifugation details (device manufacturer, type of rotor) - this is becoming important nowadays

Discussion

L245-247 -'This may indicate the importance of low saturation in the course of GERD and could explain the beneficial effect of CPAP therapy in patients without OSA who have been diagnosed with GERD [9,31-32]' - this statement and citations should be moved or added into Introduction section as a part of a hypothesis formulation.

Author Response

Thank you very much for the review and for your relevant comments.

I'am attaching the text with the corrections as suggested, marked in blue font.

Kind regards,

P.Pardak

Reviewer 2 Report

 have read with an interest this original paper of Pardak al. The authors took up an interesting topic of the associations of OSA, Obestatin, Leptin, and Ghrelin with Gastroesophageal Reflux. In my opinion, the following issues need correction:

Major:
1) In my opinion, the introduction should provide a wider outlook on sleep apnea comorbidities. The following papers may be useful in this context: 10.3389/fphys.2020.01035 and 10.1186/s40248-019-0172-9.
2) The introduction should be concluded with the hypothesis of the study.
3) The Statistical Analysis should be described in the separated section. The name of the company and its country should be provided after the name of the employed software. Did you check the normality of obtained data? Average +- SD should be used for data with normal distribution, median with IQR should be employed in other cases. p<0.001 should be stated as p<0.001 instead of exact value.
4) It may be beneficial to the paper to provide a wider outlook on the dependencies between gastroenterological diseases and sleep disorders: PMID: 27134599, 10.3390/jcm9092921, or 10.1111/nmo.13978

Minor:
1) The titles of subsections should be removed from the abstract (e.g., background, results)
2) p-values should be added to Figures 1 and 2.
3) Strengths and limitations of the study should be listed in the separated section.

Author Response

Thank you very much for the review and for your relevant comments.

 I am attaching the text with the corrections as suggested, marked in blue font.

Answers to questions about statistical analysis:

1/ Did you check the normality of obtained data?

Yes, the normality of distribution of the analyzed features was verified using the Shapiro-Wilk test. Normality plots and histograms were also used to visually assess the shape of the distribution. The results of the normality test for Ghrelin suggested very clear deviations from normality (p <0.001), and the distribution of Ghrelin values was very clearly asymmetric (skewness coefficients at the level of 3 or more). Additionally, the distribution of the Leptin measurements was not normal (p <0.05 for some of the measurements, and the skewness was approximately 0.5 or more). On the other hand, for Obestatin, p values ​​obtained with the Shapiro-Wilk test were again very low (for different hours of measurement, p <0.01 or even p <0.001). In summary, since most of the studied clinical parameters (laboratory and others) did not have a normal distribution, it was justified to use non-parametric tests. Due to length limitations of the article and the fact that it was not important for practical conclusions, results from the Shapiro-Wilk test were not stated in the paper. Only the results were mentioned along with a description of the statistical methods.

2/ Average +- SD should be used for data with normal distribution, median with IQR should be employed in other cases.

The reviewer's suggestions were followed. The tables regarding laboratory parameters did not include standard deviation, only IQR. However, the mean value is left in the tables to allow readers to compare the results with other publications, which often only provide mean values. Tables describing the distribution of variables in the entire surveyed population also include information about the minimum and maximum.

3/           p<0.001 should be stated as p<0.001 instead of exact value.

 p-values should be added to Figures 1 and 2

The reviewer's suggestions were followed.

Kind regards,

P.Pardak

Round 2

Reviewer 1 Report

All of my remarks were successfuly addressed.

Reviewer 2 Report

The authors addressed all the comments suffiicently improving the manuscript.